# Accuracy and Efficacy of Artificial Intelligence-Derived Automatic Measurements of Transthoracic Echocardiography in Routine Clinical Practice

**DOI:** 10.3390/jcm13071861

**Published:** 2024-03-24

**Authors:** Noriko Shiokawa, Masaki Izumo, Toshio Shimamura, Yui Kurosaka, Yukio Sato, Takanori Okamura, Yoshihiro Johnny Akashi

**Affiliations:** 1Ultrasound Center, St. Marianna University Hospital, 2-16-1 Sugao, Miyamae-ku, Kawasaki 216-8511, Japan; shio_no55nolife@ybb.ne.jp (N.S.); toshio.shimamura@marianna-u.ac.jp (T.S.); yui.kurosaka@marianna-u.ac.jp (Y.K.); okamura@marianna-u.ac.jp (T.O.); 2Department of Cardiology, St. Marianna University School of Medicine, 2-16-1 Sugao, Miyamae-ku, Kawasaki 216-8511, Japan; y3sato@marianna-u.ac.jp (Y.S.); johnny@marianna-u.ac.jp (Y.J.A.)

**Keywords:** transthoracic echocardiography, artificial intelligence, automatic measurement

## Abstract

**Background**: Transthoracic echocardiography (TTE) is the gold standard modality for evaluating cardiac morphology, function, and hemodynamics in clinical practice. While artificial intelligence (AI) is expected to contribute to improved accuracy and is being applied clinically, its impact on daily clinical practice has not been fully evaluated. **Methods**: We retrospectively examined 30 consecutive patients who underwent AI-equipped TTE at a single institution. All patients underwent manual and automatic measurements of TTE parameters using the AI-equipped TTE. Measurements were performed by three sonographers with varying experience levels: beginner, intermediate, and expert. **Results:** A comparison between the manual and automatic measurements assessed by the experts showed extremely high agreement in the left ventricular (LV) filling velocities (E wave: r = 0.998, A wave: r = 0.996; both *p* < 0.001). The automated measurements of LV end-diastolic and end-systolic diameters were slightly smaller (−2.41 mm and −1.19 mm) than the manual measurements, although without significant differences, and both methods showing high agreement (r = 0.942 and 0.977, both *p* < 0.001). However, LV wall thickness showed low agreement between the automated and manual measurements (septum: r = 0.670, posterior: r = 0.561; both *p* < 0.01), with automated measurements tending to be larger. Regarding interobserver variabilities, statistically significant agreement was observed among the measurements of expert, intermediate, and beginner sonographers for all the measurements. In terms of measurement time, automatic measurement significantly reduced measurement time compared to manual measurement (*p* < 0.001). **Conclusions:** This preliminary study confirms the accuracy and efficacy of AI-equipped TTE in routine clinical practice. A multicenter study with a larger sample size is warranted.

## 1. Introduction

Transthoracic echocardiography (TTE) is the most commonly utilized modality to assess cardiac morphology, function, and hemodynamics in routine clinical practice owing to its cost-effectiveness and minimally invasive nature [1,2]. TTE is essential in various scenarios, including detecting myocardial damage due to anticancer drugs [3,4] and the rising incidence of heart failure and valvular diseases in an increasingly aging society [5]. This modality is essential for determining the need for and effectiveness of treatments. Consequently, the demand for TTE has surged in recent years and is expected to continue to rise. Beyond the traditionally measured left-sided size and function, there is now a growing demand for assessing right-sided function and global longitudinal strain using speckle-tracking echocardiography [6,7,8,9,10,11,12,13], for which the number of parameters has continued to increase, increasing the complexity and time required for testing a problem in daily clinical practice. Additionally, the accuracy of echocardiographic assessments is known to be operator-dependent, with interobserver variability being a significant concern [1]. Recently developed TTE systems equipped with artificial intelligence (AI)-assisted automatic measurement capabilities promise to enhance examination accuracy by reducing both measurement time and interobserver variability [14,15,16,17]. However, the impact of these advancements on daily clinical practice has not yet been investigated. The purpose of this study was to compare the accuracy and time efficiency of routine examinations performed using an ultrasound system equipped with an AI application.

## 2. Methods

### 2.1. Study Population

This retrospective study enrolled 49 patients examined with AI-equipped echocardiography using an EPIQ CVx, X5-1c transducer (Philips Healthcare Ultrasound LLC, Bothell, WA, USA) between 15 December 2022 and 6 January 2023 at St Marianna Medical University Hospital. Of these, a total of 19 cases were excluded: atrial fibrillation (9 cases), PVC frequency (1 case), pacemaker implantation (1 case), severe bradycardia (1 case), no images recorded for analysis (6 cases), and difficult to delineate (1 case). Ultimately, 30 cases were included. The study protocol was approved by our ethics committee (approval no. 6189), and patient consent was obtained using an opt-out approach. The AI used in this study was based on TTE data collected from a large number of adults of various ethnicities in a multicenter, international approach spanning multiple continents in the Americas (North, Central, and South America), Europe, Africa, and Asia (North, Central, and South Asia), totaling more than 3000 TTE results from healthy subjects and patients with various heart disease, which were used to train and validate the algorithm.

### 2.2. Transthoracic Echocardiography

Two-dimensional (2D) and Doppler echocardiography were performed according to the American Society of Echocardiography guidelines [2,7]. Measurements included parasternal left ventricular (LV) long-axis images of interventricular septum (IVS), left ventricular posterior wall (LVPW), LV end-diastolic diameter (LVDd), LV dimension diastoles (LVDs), mitral inflow velocity (E, A, and E/A) by pulsed Doppler, deceleration time (DT), LV outflow tract (LVOT) diameter to measure time-integrated values (VTI—velocity time integral) and peak velocity, tissue Doppler measurement of septal mitral annular velocity waveforms e’ and a’, and lateral mitral annular velocities e’ and a’ using tissue Doppler. The automatic measurements were performed using an automatic analysis software designed with AI-based algorithms, for the same parameters as in the manual measurements. The measurements were carried out offline on the equipment with the data stored on a hard disk.

### 2.3. Manual Measurements

Measurements of the aforementioned parameters were performed manually on the TTE machine for LVDd, IVS, and LVPW, and three measurements were taken at the first frame immediately after mitral valve closure or at the peak of the R wave of the ECG at end-diastole, just below the mitral valve leaflet and perpendicular to the endocardial border of the ventricular septum and posterior wall. LVDs were measured when the LV was smallest just before the mitral valve opened during diastole. Mitral inflow velocities were measured from the recorded waveforms for E wave (early diastole), A wave (atrial systole) velocity, and E wave deceleration time (DT). Mitral annular velocity waveforms were measured from the recorded waveforms on the septal and lateral sides of the mitral annulus, measuring e’ (early diastolic) and a’ (atrial systolic) velocities, respectively. The LVOT velocity was traced, and the velocity–time integral (VTI) and peak velocity of the LVOT flow were measured.

### 2.4. Automatic Measurements

The automatic measurements were performed using an automatic analysis software designed with AI-based algorithms for the same parameters as in the manual measurements. After automatic measurement, corrections were made as needed (Figure 1). The “Auto Measure” function was trained to predict the measured values for all items using an algorithm in accordance with the American Society of Echocardiography Guidelines [7], and the time phase setting and measurement were performed automatically when the panel button for each measurement item was pressed. For the LVDd, IVS, LVPW, and LVDs, the time phase was automatically set to end-diastole or end-systole as appropriate. If the time phase did not match, manual correction was made so that the frame was set at end-diastole or end-systole as appropriate. Measurements of left ventricular wall thickness and left ventricular diameter were taken just below the apex of the mitral valve leaflet, perpendicular to the left ventricular long axis. The measurements were made just above the boundary between the ventricular septum and the lumen and between the left ventricular posterior wall and the pericardium, and corrections were made for measurement sites that did not align, such as in the case of poor images (Figure 1B). Doppler velocities were also automatically measured, and corrections were made for those Doppler velocities that were not measured correctly due to irregular envelopes (Figure 1D).

### 2.5. Reproducibility

The manual and automated measurements for all cases and all parameters were tested for reproducibility by three investigators. To reduce potential bias between measurements, the manual and automatic measurements were performed at least two days apart. All measurements were also performed by three sonographers with different years of echocardiographic experience: a beginner with less than one year of practice, an intermediate technician with less than five years, and an expert with more than 20 years. Interobserver and intraobserver measurement reproducibility using the manual and automatic measurements was performed in all cases. Two investigators independently analyzed the same images. These investigators were blinded to each other’s results and all other previous measurements.

### 2.6. Examination Time Analysis

Each investigator recorded the time required to take the manual and automatic measurements. The timer was paused when the reader switched between images and was restarted with the reinitiation of further measurements.

### 2.7. Statistical Analysis

Continuous variables were expressed as median and inter-quartile range (IQR) or percentage according to the data distribution. Both Spearman’s rank correlation coefficient and the Bland–Altman method were used to investigate measurement error between manual and automatic measurements and between investigators. The time required for measurement was analyzed using the Wilcoxon signed rank sum test, with *p* < 0.05 indicating a significant difference. Statistical analyses were performed using GraphPad Prism version 10.1.0 (264) for Mac OS (La Jolla, CA, USA).

## 3. Results

A summary of the baseline characteristics is provided in Table 1. Of the 30 patients studied, 16 (53%) were male and the mean age was 72 years [52.0–95.0]. The study patients included 12 with valvular disease (40%), 1 with hypertension (3%), 3 with ischemic cardiomyopathy (10%), 3 with non-ischemic cardiomyopathy (10%), 2 with pulmonary hypertension (7%), and 9 with normal LVEF (30%) with a LVEF of 61.5% [35–78]. Image quality was good in 12 (40%), fair in 9 (30%), and poor in 9 (30%).

### 3.1. Comparisons of Manual vs. Automatic Measurements

Both the measurements manually evaluated by the expert investigator and the AI-based automatic measurements are shown in Table 2. For LVDd and LVDs, the automatic measurements showed slightly smaller values than the manual measurements and were consistent (LVDd: r = 0.942; LVDs: r = 0. 977) (both *p* < 0.001), with a bias of −2.41 mm for LVDd and −1.19 mm for LVDs, according to the Bland–Altman analysis. There was little substantial dissimilarity between the two measurements. On the other hand, the correlation coefficients for IVS and LVPW were slightly lower (IVS: r = 0.670; LVPW: r = 0.561) and automatic measurements tended to measure a slightly thicker wall thickness than manual measurements. For the E and A waves of the LV inflow velocity, a very high agreement was observed between the automatic and manual measurements (E wave: r = 0.998; A wave: r = 0.996), with a small bias, based on the Bland–Altman analysis (E wave: 1.37 cm/s; A wave: 0.08 cm/s). For DT (deceleration time) and tissue Doppler (e’ and a’ waves), automated measurements also showed high agreement with the manual measurements (correlation coefficient > 0.832), especially for e’ in the lateral wall (r ≥ 0.957 or higher), which was in very strong agreement with the manual measurements. The LVOT VTI and peak velocity also showed very high agreement between the two measurements (VTI: r = 0.982; peak velocity: r = 0.972), and the bias from the Bland–Altman analysis was very small (VTI: −0.13 cm; peak velocity: 2.87 cm/s). In addition, Table 3 shows the results of a study on the impact of image quality on measurement accuracy. Better image quality improves the accuracy of automatic measurement.

### 3.2. Reproducibility

A comparison between the manual measurements taken by experts and those taken by intermediate users and beginners is shown in Table 4. The correlation coefficients ranged from 0.549 to 0.992, indicating that most of the indexes were reliable for the measurement of each parameter, regardless of the experience level. Furthermore, the manual measurements by experts and the automatic measurements by intermediate users and beginners are shown in Table 5. Overall, the correlation coefficients were high, with *p* < 0.05 for all measurements, indicating statistically significant agreement between the intermediate and beginner automatic measurements and the expert manual measurements. A high level of agreement in LVDd and LVDs was observed between the manual (expert) and automatic (intermediate and beginner) measurements, with a correlation coefficient of 0.92 for each. For LV wall thickness, the agreement was lower for IVS (intermediate users: r = 0.70; beginners: r = 0.75) and for LVPW (intermediate users: r = 0.51; beginners: r = 0.41) than for the other measurements. High agreement was found for VTI and peak velocity in the LVOT (beginners: r = 0.97; intermediate users: r = 0.95). The E and A waves also showed very high agreement (beginners: r = 0.99; intermediate users: r = 0.99). Automatic measurements tended to be more consistent than manual measurements for most measurements.

### 3.3. Examination Duration

Figure 2 show the results of a comparison of the time required for manual and automatic measurements for investigators with different levels of experience. Experts significantly reduced their measurement time with the use of automatic measurement (manual (81.5 [73.4–92.0] seconds) vs. automatic (59.0 [38.0–75.0] seconds; *p* < 0.001). No statistically significant differences were found between the two intermediate groups (manual (80.0 [76.0–99.5] seconds) vs. automatic (82.0 [66.3–95.0] seconds); *p* = 0.296). Similar to the experts, beginners took significantly less time with automatic measurement (manual (121.5 [103.0–169.3] seconds) vs. automatic (89.0 [73.0–103.3] seconds; *p* < 0.001). Table 6 shows the results of a comparison of the variation in examination duration between manual and automatic measurements according to image quality. Beginners consistently took longer than intermediate users and experts for all image qualities in manual measurements, with a median of 120.0 [105.0–205.5] seconds for poor image quality, 121.5 [103.8–169.3] ms for fair image quality, and 124.5 [104.3–146.8] ms for good image quality in manual measurements. Automatic measurements took less time than manual measurements: poor image quality, 84 [72–105] s; fair image quality, 97 [79–114] s; and good image quality, 89 [67–96] s. For intermediate users, automatic measurements took a slightly longer time than manual measurements for poor and fair image quality. However, for good image quality, automatic measurements took significantly less time than manual measurements. Among the experts, automatic measurements reduced the measurement time for all image qualities compared to manual measurements, and similar to the intermediate level, as the image quality improved, so did the measurement time with automatic measurements (poor image quality, 83.0 [55.5–95.0] s; fair image quality, 55.0 [49.0–78.0] s; and good image quality, 52.3 [41.5–67.8] s).

## 4. Discussion

This study is the first to investigate the accuracy of commercially available AI-assisted automated TTE measurements and their implications for routine clinical practice.

The key findings are as follows:(1)High accuracy of AI: The accuracy of many echocardiographic parameters was high for automated measurements using AI. This was particularly true for the Doppler echocardiography, which showed a high degree of agreement with manual measurements.(2)Reduced examination time required: Automatic measurement reduced the examination time compared to manual measurement, suggesting that it may contribute to increased efficiency of the examination. The reduction was particularly noticeable for beginners.(3)Reduction in interobserver variabilities: The use of automated measurements reduced interobserver variabilities between experts and beginners, indicating that it can also be used as an educational tool.

Previous studies related to AI-based automation in echocardiography have reported the automation of morphological and functional assessments and the use of machine learning algorithms for image recognition and analyses for use in diagnosis [18,19,20,21,22,23,24,25,26]. This study focused on basic measurement parameters performed in routine clinical practice. AI-based automated measurements showed high agreement with conventional manual measurements for several measurements performed in echocardiography, with particularly significant agreement (r > 0.99) for Doppler indexes. This is a result of AI technology facilitating the standardization of measurements, indicating that AI facilitates measurement standardization and reduces interobserver variabilities. However, the results for LV wall thickness (IVS and LVPW) were less consistent than those for the other measurements, and care should be taken to ensure that the measurement of LV wall thickness does not include the wall column, the right ventricular zone, or the subvalvular tissue of the tricuspid valve and that the boundary between the right and left ventricular cavity is measured so that the LV posterior wall side is not included, nor the boundary between the LV cavity and the myocardium or the mitral valve subvalvular tissue [2]. This is thought to be due to the fact that measurements require care and are susceptible to influence, such as not including the boundary between the left ventricular lumen and myocardium or the subvalvular tissue of the mitral valve, indicating the need for careful assessment as appropriate in certain parameters.

With regard to the time required to carry out multi-item measurements, the results indicate that automatic measurements have the potential to reduce measurement time. For both expert and beginner groups, the use of automatic measurements resulted in a significant reduction in the time required to take measurements. This suggests that automated measurement is an effective tool to improve the efficiency of ultrasound examinations not only for technicians with advanced expertise, but also for less-experienced technicians. In a previous study, Knackstedt et al. [15] found that fully automated LV volume and ejection fraction measurements reduced the measurement time and enabled more efficient examinations to be performed. In the present study, a similar reduction in measurement time was achieved. The fact that experts were able to reduce the time required for measurement by using automated measurement complements a great deal of experience and skill, in addition to the inspection itself being carried out more quickly. It is clear that beginners can significantly reduce the time required to take measurements by using automatic measurement. The results suggest that beginners may have taken longer to perform manual measurements due to uncertainty and technical inexperience. Although echocardiographic studies using AI have reported its usefulness as an image acquisition guide for inexperienced beginners and as an educational and diagnostic aid [14,27], the automatic measurement used in this study may also be useful for beginners. The automatic measurement used in this study can also be used by beginners as a guide for automatic analysis itself, which may allow for a faster examination and more efficient skills training. In contrast, no statistically significant difference in the time required for measurement was observed for intermediate users; however, when the time required was divided into groups according to image quality, a statistically significant difference was observed between automatic and manual measurements as image quality improved, indicating that measurement can be carried out more efficiently. This suggests that the AI-based automatic measurement algorithm works efficiently when the image quality is good, but requires more correction and time when image quality is low. The lack of statistically significant differences between the automatic and manual measurements performed by intermediate users may be due to the possibility that they have sufficient experience in echocardiography to have a degree of proficiency in manual measurements but have not adapted to automatic measurement techniques, which may make it difficult for the benefits of AI to emerge. Additional research is needed on this point, in particular with technicians of many different levels of expertise. However, taken together, the results on the use of automatic measurement in beginners indicate that automatic measurement may significantly improve the measurement time and accuracy of beginners compared to manual measurement, even for all image qualities. The improvements were particularly noticeable for poor images, suggesting that automatic measurement can be used by beginners as an inspection aid tool to enable reliable measurements, as well as an educational support tool when learning the technique. While the current study focused on the impact of AI on the accuracy of echocardiographic measurements and examination time, future studies should investigate how AI measurements can influence diagnostic flow and decision making in clinical practice.

### Study Limitations

The study had several limitations. First, this study is a small, single-center study with limited data, which may be insufficient for determining statistical significance. There may also be a lack of data coverage and diversity of patients with different cases and clinical backgrounds. Secondly, the assessment of image quality includes subjective assessments, which may lead to bias. Thirdly, although comparisons with other modalities, such as MRI, have not been made, the automatic measurement parameters in this study are not volume data of the LV and their usefulness in routine clinical practice has been verified, which is not necessarily. Finally, this study has not been compared with AI systems from other vendors, nor has it examined the superiority of our system’s AI relative to others, warranting further investigation.

## 5. Conclusions

In this study, AI-based automated measurements were found to have the potential to achieve high accuracy in routine clinical practice. The results also suggest that these types of measurements could contribute to reducing examination duration and eliminating interobserver variabilities. Although large-scale multicenter prospective studies are warranted in the future to confirm and expand on these findings, it is expected that AI-equipped TTE will be widely used in daily clinical practice.

## Figures and Tables

**Figure 1 jcm-13-01861-f001:**
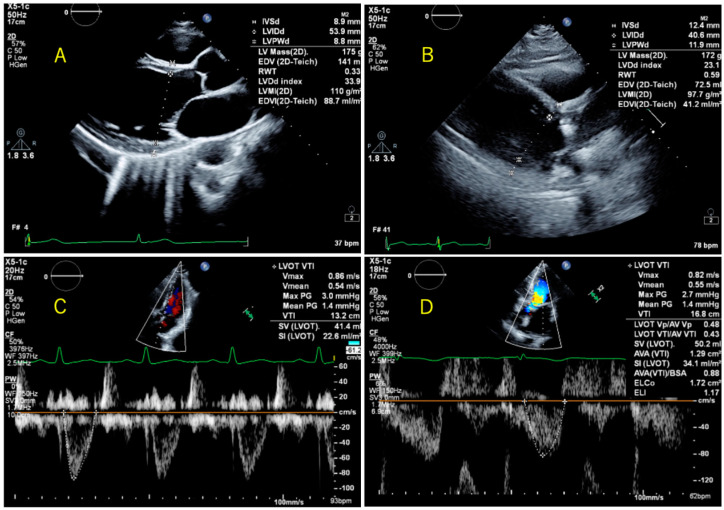
Representative cases where fully automatic measurement was possible (**A**,**C**), and cases where correction was necessary (**B**,**D**). (**A**) had good image quality and did not need to be corrected after automatic measurement. (**B**) was of poor image quality, and the measurement position did not capture the boundaries of the left ventricle, so a correction was made. (**C**) No correction was made after automatic measurement. (**D**) Corrections were made because the boundaries of the pulsed Doppler waveform were not captured.

**Figure 2 jcm-13-01861-f002:**
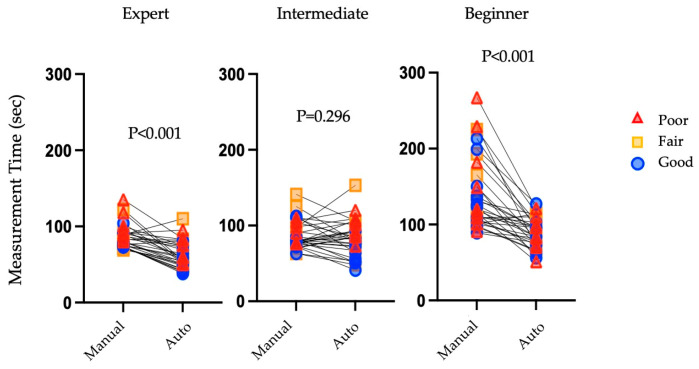
Comparisons of examination duration in each level of investigators. Experts and beginners showed a reduction in examination time for automatic measurement compared to manual measurement, but no difference in examination time was observed between the two methods for intermediate users.

**Table 1 jcm-13-01861-t001:** Baseline characteristics.

	N = 30
Men	16 (53%)
Age, year	72 [52.0–78.5]
Body surface area, m^2^	1.61 [1.44–1.81]
HR (beats/min)	64.5 [56.8–78.0]
Systolic blood pressure, mmHg	130 [121.3–142.3]
Diastolic blood pressure, mmHg	71.5 [63.8–79.8]
Primary diagnosis	
Valvular heart disease	12 (40%)
Hypertensive heart disease	1 (3%)
Ischemic cardiomyopathy	3 (10%)
Non-ischemic cardiomyopathy	3 (10%)
Pulmonary hypertension	2 (7%)
other	9 (30%)
Echocardiographic measurements	
LVEDV, mL	92 [75.8–140.3]
LVESV, mL	33.5 [28.5–58.3]
LVEF, %	61.5 [55.8–68.0]
LAVi, mL/m^2^	34.0 [25.7–48.8]

LVEDV: left ventricular end-diastolic volume; LVESV: left ventricular end-systolic volume; LVEF: left ventricular ejection fraction; LAVi: left atrial volume index.

**Table 2 jcm-13-01861-t002:** Comparison of manual vs. automatic measurements assessed by expert.

	Manual (IQR)	Auto (IQR)	r	*p* Value	Bland–Altman	
					Bias	Difference
LVDd, mm	48.5 [44.5–50.7]	45.2 [42.9–48.2]	0.94	<0.001	−2.41	−5.95–1.14
LVDs, mm	31.1 [26.7–37.3]	28.5 [26.6–34.9]	0.98	<0.001	−1.19	−4.18–1.80
IVS, mm	8.6 [7.9–10.5]	9.2 [8.5–10.7]	0.67	<0.001	0.33	−2.89–3.55
LVPW, mm	8.7 [8.0–10.5]	9.8 [8.9–11.6]	0.561	0.0013	0.87	−2.44–4.18
E velocity, cm/s	78.2 [56.2–105.0]	83.2 [55.7–105.5]	0.998	<0.001	1.37	−3.28–6.01
A velocity, cm/s	72.9 [53.7–95.1]	72.1 [52.8–96.0]	0.996	<0.001	0.08	−4.90–5.07
DT, ms	193.5 [163.0–237.8]	195.5 [175.8–225.5]	0.832	<0.001	1.05	−60.2–62.3
e’ velocity (sep), cm/s	6.0 [5.1–7.9]	6.3 [5.4–8.5]	0.957	<0.001	0.25	−0.85–1.36
a’ velocity (sep), cm/s	9.3 [7.0–10.6]	9.4 [7.0–11.2]	0.976	<0.001	0.19	−0.91–1.28
e’ velocity (lat), cm/s	7.7 [6.4–10.4]	8.0 [6.5–9.9]	0.974	<0.001	0.17	−1.29–1.64
a’ velocity (lat), cm/s	9.9 [8.7–12.2]	9.9 [8.9–11.9]	0.982	<0.001	0.003	−1.15–1.16
LVOT VTI, cm	17.7 [15.1–22.5]	17.6 [15.2–22.3]	0.982	<0.001	−0.13	−1.86–1.60
LVOT peak velocity, cm/s	81.0 [71.0–90.5]	85.0 [ 72.5–96.0]	0.972	<0.001	2.87	−4.64–10.4

LVDd, left ventricular end-diastolic diameter; LVDs, left ventricular end-systolic diameters; IVC, interventricular septum; LVPW, left ventricular posterior wall; DT, deceleration time; LVOT, left ventricular outflow tract.

**Table 3 jcm-13-01861-t003:** Comparison of manual vs. automatic measurements based on the image quality.

	Manual (IQR)	Auto (IQR)	r	*p* Value	Bland–Altman	
					Bais	Difference
**Poor image**						
LVDd, mm	46.9 [43.2–48.5]	43.1 [42.7–46.0]	0.76	<0.001	−2.35	−6.73–2.23
LVDs, mm	30.9 [26.0–32.3]	38.7 [24.4–32.9]	0.92	0.104	−1.06	−6.89–4.78
IVS, mm	8.9 [8.4–9.8]	9.5 [8.8–10.7]	−0.20	0.013	0.76	−2.15–3.68
LVPW, mm	9.1 [8.1–1-.4]	10.1 [9.6–10.8]	0.01	0.204	1.13	−2.42–4.69
E velocity, cm/s	61.3 [48.2–110]	62.9 [53.0–66.0]	031	0.003	0.25	−6.91–7.41
A velocity, cm/s	87.5 [57.5–95.7]	84.9 [48.7–110]	0.98	0.192	−0.97	−8.88–6.95
DT, msec	230 [169–280]	214 [185–256]	0.47	0.789	−5.65	−88.3–77.0
e’ velocity (sep), cm/s	5.8 [5.1–7.3]	6.0 [5.8–6.5]	0.70	<0.001	0.44	−1.15–2.03
a’ velocity (sep), cm/s	8.9 [6.7–10.6]	9.1 [7.1–12.1]	0.91	0.166	0.34	−2.04–2.73
e’ velocity (lat), cm/s	6.9 [5.9–8.6]	6.6 [6.3–7.9]	0.87	0.232	0.11	−1.42–1.64
a’ velocity (lat), cm/s	9.6 [8.4–12.6]	9.8 [9.2–12.0]	0.95	0.327	0.14	−1.53–1.82
LVOT VTI, cm	17.5 [12.6–20.7]	17.5 [13.6–20.2]	0.95	0.421	0.20	−2.06–2.47
LVOT peak velocity, cm/s	82.0 [58.0–91.0]	86.0 [61.0–99.0]	0.87	0.003	2.93	−6.27–12.1
**Fair image**						
LVDd, mm	49.3 [47.4–52.5]	47.9 [45.2–50.2]	0.79	0.033	−1.01	−6.01–3.99
LVDs, mm	31.9 [27.7–38.2]	31.8 [27.7–36.2]	0.91	0.057	−0.91	−6.55–4.73
IVS, mm	8.2 [6.7–11.3]	8.7 [8.2–12.6]	0.77	0.030	0.57	−2.63–3.78
LVPW, mm	8.7 [7.8–9.6]	8.9 [8.7–10.0]	0.66	0.309	0.36	−2.25–2.98
E velocity, cm/s	71.8 [56.6–99.6]	74.9 [60.3–104]	0.99	<0.001	−1.89	−5.58–1.81
A velocity, cm/s	89.1 [54.5–104]	87.7 [55.3–102]	0.97	0.183	−0.93	−7.65–5.80
DT, msec	203 [153–219]	198 [180–227]	0.66	0.118	8.26	−83.1–99.8
e’ velocity (sep), cm/s	5.6 [4.1–6.3]	5.8 [4.1–6.5]	0.93	0.133	0.15	−0.82–1.12
a’ velocity (sep), cm/s	9.6 [5.8–11.0]	10.0 [5.8–11.2]	0.99	0.268	0.12	−0.93–1.16
e’ velocity (lat), cm/s	7.3 [5.4–8.7]	7.7 [7.3–8.9]	0.93	0.023	0.36	−1.41–2.12
a’ velocity (lat), cm/s	10.2 [7.9–12.0]	10.3 [8.9–11.4]	0.97	0.392	0.11	−1.01–1.24
LVOT VTI, cm	17.2 [15.0–22.3]	17.6 [14.7–22.2]	0.94	0.702	0.23	−2.61–3.07
LVOT peak velocity, cm/s	81.0 [70.0–89.0]	84.0 [72.0–96.0]	0.93	0.003	2.33	−4.75–9.42
**Good image**						
LVDd, mm	46.5 [43.4–53.5]	44.6 [40.8–53.2]	0.96	0.004	−0.96	−4.69–2.76
LVDs, mm	28.5 [25.9–34.4]	27.7 [27.0–31.0]	0.89	0.071	−0.62	−4.10–2.86
IVS, mm	9.2 [7.3–10.6]	9.3 [8.0–10.6]	0.77	0.189	0.30	−2.23–2.83
LVPW, mm	9.6 [8.1–10.8]	9.5 [9.0–11.1]	0.64	0.743	0.04	−2.74–2.83
E velocity, cm/s	90.2 [64.5–99.2]	93.8 [65.5–98.8]	0.98	<0.001	2.44	−3.38–8.26
A velocity, cm/s	62.9 [40.7–71.3]	64.0 [41.4–73.5]	0.98	0.021	1.52	−5.09–8.13
DT, msec	190 [165–220]	187 [173–211]	0.75	0.638	1.93	−44.3–48.1
e’ velocity (sep), cm/s	7.3 [5.2–9.0]	7.9 [6.0–9.2]	0.97	<0.001	0.51	−0.52–1.54
a’ velocity (sep), cm/s	9.2 [7.7–10.2]	9.4 [7.8–10.2]	0.95	<0.001	0.33	−9.43–1.09
e’ velocity (lat), cm/s	8.2 [6.8–10.3]	9.2 [7.2–11.0]	0.96	<0.001	0.64	−1.10–2.38
a’ velocity (lat), cm/s	9.3 [8.7–11.7]	9.7 [8.9–11.5]	0.88	0.117	−0.06	−2.96–2.85
LVOT VTI, cm	16.7 [15.3–21.1]	17.1 [15.6–21.0]	0.91	0083	0.34	−1.78–2.44
LVOT peak velocity, cm/s	78.5 [74.3–90.3]	80.0 [75.3–94.3]	0.93	<0.001	3.03	−5.15–11.2

LVDd, left ventricular end-diastolic diameter; LVDs, left ventricular end-systolic diameters; IVC, interventricular septum; LVPW, left ventricular posterior wall; DT, deceleration time; LVOT, left ventricular outflow tract.

**Table 4 jcm-13-01861-t004:** Comparison of manual measurements among experts, intermediate users, and beginners.

		Intermediate					Beginner				
	Expert (IQR)	IQR	r	*p* Value	Bland–Altman		IQR	r	*p* Value	Bland–Altman	
					Bias	Difference				Bias	Difference
LVDd, mm	48.5 [44.5–50.7]	47.9 [43.6–50.1]	0.90	<0.001	−0.78	−3.78–5.35	46.7 [43.5–49.1]	0.88	<0.001	−1.73	−5.64–2.18
LVDs, mm	31.1 [26.7–37.3]	29.3 [25.8–36.1]	0.89	<0.001	−1.36	−7.04–4.32	28.7 [26.3–35.5]	0.88	<0.001	−1.25	−5.70–3.20
IVS, mm	8.6 [7.9–10.5]	9.4 [8.1–10.5]	0.66	<0.001	0.18	−2.63–2.99	8.6 [7.4–9.3]	0.72	<0.001	−0.66	−3.45–2.13
LVPW, mm	8.7 [8.0–10.5]	9.3 [8.5–9.3]	0.55	<0.01	0.54	−2.22–3.28	9.4 [8.0–10.3]	0.70	<0.001	0.53	−2.60–3.67
E velocity, cm/s	78.2 [56.2–105.0]	79.3 [54.5–100]	0.99	<0.001	−2.15	−7.15–3.36	79.9 [58.3–104]	0.98	<0.001	0.32	−5.78–6.15
A velocity, cm/s	72.9 [53.7–95.1]	69.7 [49.2–93.4]	0.99	<0.001	−3.13	−9.38–3.12	74.2 [55.0–95.5]	0.98	<0.001	1.45	−5.75–8.66
DT, msec	193.5 [163.0–237.8]	186 [140–233]	0.89	<0.001	−11.6	−66.8–43.5	209 [178–232]	0.78	<0.001	9.88	−41.7–61.5
e’ velocity (sep), cm/s	6.0 [5.1–7.9]	5.8 [4.9–7.3]	0.93	<0.001	−0.51	−1.66–0.64	6.0 [4.9–7.7]	0.92	<0.001	−0.01	−1.21–1.18
a’ velocity (lat), cm/s	9.3 [7.0–10.6]	9.0 [6.6–10.5]	0.97	<0.001	−0.23	−1.30–0.84	8.9 [7.2–10.8]	0.92	<0.001	0.06	−1.59–1.71
e’ velocity (lat), cm/s	7.7 [6.4–10.4]	7.0 [5.5–9.4]	0.89	<0.001	−0.82	−2.75–1.12	7.9 [6.3–10.4]	0.92	<0.001	−0.13	−1.25–1.60
a’ velocity (lat), cm/s	9.9 [8.7–12.2]	9.6 [8.5–12.3]	0.85	<0.001	−0.04	−3.65–3.57	9.6 [8.4–11.8]	0.96	<0.001	−0.14	−1.61–1.33
LVOT VTI, cm	17.7 [15.1–22.5]	16.0 [14.9–21.3]	0.94	<0.001	−1.17	−3.50–1.16	16.7 [14.8–21.1]	0.95	<0.001	−0.88	−3.24–1.48
LVOT peak velocity, cm/s	81.0 [71.0–90.5]	79.5 [70.5–90.0]	0.96	<0.001	−0.57	−8.11–6.98	82.5 [70.8–91.0]	0.95	<0.001	0.90	−8.53–10.3

LVDd, left ventricular end-diastolic diameter; LVDs, left ventricular end-systolic diameters; IVC, interventricular septum; LVPW, left ventricular posterior wall; DT, deceleration time; LVOT, left ventricular outflow tract.

**Table 5 jcm-13-01861-t005:** Manual measurement (expert) vs. automatic measurements (intermediate and beginner).

		vs. Intermediate					vs. Beginner				
	Expert (IQR)	IQR	r	*p*-Value	Bland–Altman		IQR	r	*p*-Value	Bland–Altman	
					Bias	Difference				Bias	Difference
LVDd, mm	48.5 [44.5–50.7]	45.8 [43.0–50.1]	0.92	<0.001	−1.68	−5.72–2.56	45.1 [42.5–49.2]	0.92	<0.001	−2.52	−6.80–1.77
LVDs, mm	31.1 [26.7–37.3]	28.9 [27.1–33.2]	0.89	<0.001	−1.64	−7.85–4.57	28.6 [24.7–32.5]	0.91	<0.001	−2.30	−8.17–3.57
IVS, mm	8.6 [7.9–10.5]	9.2 [8.5–10.5]	0.70	<0.001	0.21	−2.86–3.27	8.6 [5.8–14.5]	0.75	<0.001	0.54	−2.38–3.46
LVPW, mm	8.7 [8.0–10.5]	9.6 [8.9–10.3]	0.51	0.004	0.61	−2.55–3.77	9.8 [8.9–11.4]	0.41	0.024	0.98	−2.84–4.80
E velocity, cm/s	78.2 [56.2–105.0]	83.2 [58.1–102.0]	0.99	<0.001	0.43	−6.62–7.48	83.2 [58.1–105.5]	0.99	<0.001	1.22	−4.28–6.71
A velocity, cm/s	72.9 [53.7–95.1]	72.1 [53.4–93.4]	0.99	<0.001	−0.52	−8.18–7.13	72.1 [54.3–95.7]	0.99	<0.001	0.55	−6.36–7.46
DT, msec	193.5 [163.0–237.8]	199.5 [177.0–224.8]	0.89	<0.001	0.97	−46.2–48.1	199.5 [177.0–226.0]	0.70	<0.001	0.91	−75.3–77.1
e’ velocity (sep), cm/s	6.0 [5.1–7.9]	6.1 [5.4–7.9]	0.96	<0.001	0.07	−0.98–1.12	6.4 [5.3–8.5]	0.95	<0.001	0.30	−0.10–1.50
a’ velocity (lat), cm/s	9.3 [7.0–10.6]	9.4 [7.1–11.0]	0.98	<0.001	0.22	−0.87–1.31	9.3 [7.1–11.1]	0.97	<0.001	0.23	−0.98–1.44
e’ velocity (lat), cm/s	7.7 [6.4–10.4]	8.0 [6.5–9.9]	0.98	<0.001	0.10	−1.24–1.44	8.0 [6.5–9.9]	0.97	<0.001	0.22	−1.42–1.86
a’ velocity (lat), cm/s	9.9 [8.7–12.2]	9.9 [8.9–11.9]	0.98	<0.001	−0.05	−1.26–1.17	9.9 [8.9–11.9]	0.98	<0.001	0.03	−1.11–1.17
LVOT VTI, cm	17.7 [15.1–22.5]	17.0 [15.0–20.9]	0.90	<0.001	−0.95	−4.39–2.48	17.4 [15.0–22.6]	0.96	<0.001	−0.17	−2.42–2.07
LVOT peak velocity, cm/s	81.0 [71.0–90.5]	80.0 [73.0–95.3]	0.95	<0.001	1.30	−7.36–9.96	84.0 [73.0–99.0]	0.97	<0.001	4,52	−3.50–12.6

LVDd, left ventricular end-diastolic diameter; LVDs, left ventricular end-systolic diameters; IVC, interventricular septum; LVPW, left ventricular posterior wall; DT, deceleration time; LVOT, left ventricular outflow tract.

**Table 6 jcm-13-01861-t006:** Impact of image quality on measurement time.

		Manual	Auto	*p*-Value
Beginner	Poor	120.0 [105.0–205.5]	84.0 [71.5–105.0]	0.008
	Fair	121.5 [103.8–169.3]	89.0 [73.0–103.3]	0.012
	Good	124.5 [104.3–146.8]	89.0 [67.0–96.0]	<0.001
Intermediate	Poor	90.0 [77.0–105.0]	101.0 [86.5–108.0]	0.188
	Fair	80.0 [76.0–99.5]	82.0 [66.3–95.0]	0.934
	Good	80.0 [74.3–84.8]	68.0 [51.5–78.3]	0.009
Expert	Poor	91.0 [85.0–109.0]	83.0 [55.5–95.0]	0.012
	Fair	75.0 [71.5–90.5]	55.0 [49.0–78.0]	0.047
	Good	76.0 [73.3–88.3]	52.5 [41.5–67.8]	<0.001
All	Poor	100 [87.0–118.0]	84.0 [73.0–101.0]	0.004
	Fair	92.0 [72.0–122.0]	81.0 [61.0–104.0]	0.003
	Good	84.5 [76.0–110.0]	67.0 [52.3–82.5]	<0.001

## Data Availability

On reasonable request, derived data supporting the findings of this study are available from the corresponding author after approval from the Ethical.

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
