# Peer review of "Accuracy and Efficacy of Artificial Intelligence-Derived Automatic Measurements of Transthoracic Echocardiography in Routine Clinical Practice"

_jcm, 2024, doi:10.3390/jcm13071861_

Round 1

Reviewer 1 Report

Comments and Suggestions for Authors

This is a manuscript  of great interest and relevance given the growing rise of artificial intelligence and its progressive application in daily clinical practice.

Although the number of patients is a case in point since 19 of the initial 49 were excluded, the work is significant due to the interest aroused by the topic discussed. It is well structured and designed and the tables collect relevant information well.

The discussion is very relevant, highlighting the three main messages: precision of artificial intelligence, reducing measurement time and reducing interobserver variability.

The authors acknowledge the limitations of the study and the lack of evaluation with another more precise technique.

It should be corrected or eliminated in tjh introduction  that transthoracic echocardiography is the gold standard, since it is really cardio resonance. Although it is true that the echocardiogram is less expensive and more affordable, a comment that can be included

Author Response

We thank the Editors and Reviewers for the careful review and helpful suggestions, as well as the opportunity to submit a revised version of our manuscript. Please find below our point-by-point responses to the questions raised by the Reviewers. 

Reviewer #1

  1. It should be corrected or eliminated in tjh introduction  that transthoracic echocardiography is the gold standard, since it is really cardio resonance. Although it is true that the echocardiogram is less expensive and more affordable, a comment that can be included.

We appreciate the insightful comment from Reviewer #1. In light of your feedback, we have revised the statement as follows (line 51); ‘Transthoracic echocardiography (TTE) is the most commonly utilized modality to assess cardiac morphology, function, and hemodynamics in routine clinical practice owing to its cost-effectiveness and minimally invasive nature.’

Reviewer 2 Report

Comments and Suggestions for Authors

The manuscript titled "Accuracy and efficacy of artificial intelligence-derived automatic measurements of transthoracic echocardiography in routine clinical practice" presents an investigation into the application of Artificial Intelligence (AI) in enhancing the accuracy and efficiency of Transthoracic Echocardiography (TTE) measurements. The study addresses a significant area in cardiology, leveraging AI to potentially improve TTE measurements' accuracy and efficiency. Given the increasing reliance on TTE for cardiac assessments and the variability associated with manual measurements, this research is both relevant and timely.

There are some issues that may be improved:

- Study Population: The sample size of 30 patients, though adequate for a preliminary study, is relatively small for definitive conclusions about the broader applicability of AI in clinical practice. 

- Image Quality Consideration: The study indicates that AI's efficiency in automatic measurements may be influenced by image quality. It would be beneficial to provide a more detailed analysis of how image quality affects AI's performance.

- Comparative Analysis: While the study focuses on AI's accuracy and efficiency in automatic measurements, a comparison with other AI systems or algorithms would provide valuable insights into the relative performance of the system used in this study.

- Clinical Impact Discussion: The discussion could be expanded to explore the potential clinical implications of implementing AI in routine TTE measurements more comprehensively. Specifically, it would be helpful to consider the impact on diagnostic timelines and the potential for AI to support decision-making in clinical settings.

Comments on the Quality of English Language

The manuscript is well-written with a high level of English proficiency. However, minor revisions for grammatical precision and clarity could further polish the text.

Author Response

We thank the Editors and Reviewers for the careful review and helpful suggestions, as well as the opportunity to submit a revised version of our manuscript. Please find below our point-by-point responses to the questions raised by the Reviewers. 

Reviewer #2

  1. The sample size of 30 patients, though adequate for a preliminary study, is relatively small for definitive conclusions about the broader applicability of AI in clinical practice.

We thank Reviewer #2 for the insightful comment. We concur and have accordingly revised our conclusion to better reflect the preliminary nature of this study (line 39): ‘This preliminary study confirms the accuracy and efficacy of AI-equipped TTE in routine clinical practice. A multi-center study with a larger sample size is warranted’

  1. Image Quality Consideration: The study indicates that AI's efficiency in automatic measurements may be influenced by image quality. It would be beneficial to provide a more detailed analysis of how image quality affects AI's performance.

Thank you for emphasizing the importance of analyzing how image quality affects AI's performance in automatic measurements. We have expanded our analysis to more comprehensively assess this impact. The enhanced findings are now included in the results section, illustrated by a table 3 that delineates the verified accuracy at varying levels of image quality. Consistent with your suggestion, we observed that superior image quality significantly enhances the accuracy of automatic measurements. These updated results have been added to the results section. Furthermore, Table 6 has been revised to encompass the average examination time for all observers, thereby providing deeper insights into the study's parameters.

Manual

(IQR)

Auto

(IQR)

r

P value

Bland-Altman

Bais

Difference

Poor image

LVDd, mm

46.9[43.2–48.5]

43.1[42.7–46.0]

0.76

<0.001

-2.35

-6.73–2.23

LVDs, mm

30.9[26.0–32.3]

38.7[24.4–32.9]

0.92

0.104

-1.06

-6.89–4.78

IVS, mm

8.9[8.4–9.8]

9.5[8.8–10.7]

-0.20

0.013

0.76

-2.15–3.68

LVPW, mm

9.1[8.1–1-.4]

10.1[9.6–10.8]

0.01

0.204

1.13

-2.42–4.69

E velocity, cm/sec

61.3[48.2–110]

62.9[53.0–66.0]

031

0.003

0.25

-6.91–7.41

A velocity, cm/sec

87.5[57.5–95.7]

84.9[48.7–110]

0.98

0.192

-0.97

-8.88–6.95

DT, msec

230[169–280]

214[185–256]

0.47

0.789

-5.65

-88.3–77.0

e’ velocity (sep), cm/sec

5.8[5.1–7.3]

6.0[5.8–6.5]

0.70

<0.001

0.44

-1.15–2.03

a’ velocity (sep), cm/sec

8.9[6.7–10.6]

9.1[7.1–12.1]

0.91

0.166

0.34

-2.04–2.73

e’ velocity (lat), cm/sec

6.9[5.9–8.6]

6.6[6.3–7.9]

0.87

0.232

0.11

-1.42–1.64

a’ velocity (lat), cm/sec

9.6[8.4–12.6]

9.8[9.2–12.0]

0.95

0.327

0.14

-1.53–1.82

LVOT VTI, cm

17.5[12.6–20.7]

17.5[13.6–20.2]

0.95

0.421

0.20

-2.06–2.47

LVOT peak velocity, cm/sec

82.0[58.0–91.0]

86.0[61.0–99.0]

0.87

0.003

2.93

-6.27–12.1

Fair image

LVDd, mm

49.3[47.4–52.5]

47.9[45.2–50.2]

0.79

0.033

-1.01

-6.01–3.99

LVDs, mm

31.9[27.7–38.2]

31.8[27.7–36.2]

0.91

0.057

-0.91

-6.55–4.73

IVS, mm

8.2[6.7–11.3]

8.7[8.2–12.6]

0.77

0.030

0.57

-2.63–3.78

LVPW, mm

8.7[7.8–9.6]

8.9[8.7–10.0]

0.66

0.309

0.36

-2.25–2.98

E velocity, cm/sec

71.8[56.6–99.6]

74.9[60.3–104]

0.99

<0.001

-1.89

-5.58–1.81

A velocity, cm/sec

89.1[54.5–104]

87.7[55.3–102]

0.97

0.183

-0.93

-7.65–5.80

DT, msec

203[153–219]

198[180–227]

0.66

0.118

8.26

-83.1–99.8

e’ velocity (sep), cm/sec

5.6[4.1–6.3]

5.8[4.1–6.5]

0.93

0.133

0.15

-0.82–1.12

a’ velocity (sep), cm/sec

9.6[5.8–11.0]

10.0[5.8–11.2]

0.99

0.268

0.12

-0.93–1.16

e’ velocity (lat), cm/sec

7.3[5.4–8.7]

7.7[7.3–8.9]

0.93

0.023

0.36

-1.41–2.12

a’ velocity (lat), cm/sec

10.2[7.9–12.0]

10.3[8.9–11.4]

0.97

0.392

0.11

-1.01–1.24

LVOT VTI, cm

17.2[15.0–22.3]

17.6[14.7–22.2]

0.94

0.702

0.23

-2.61–3.07

LVOT peak velocity, cm/sec

81.0[70.0–89.0]

84.0[72.0–96.0]

0.93

0.003

2.33

-4.75–9.42

Good image

LVDd, mm

46.5[43.4–53.5]

44.6[40.8–53.2]

0.96

0.004

-0.96

-4.69–2.76

LVDs, mm

28.5[25.9–34.4]

27.7[27.0–31.0]

0.89

0.071

-0.62

-4.10–2.86

IVS, mm

9.2[7.3–10.6]

9.3[8.0–10.6]

0.77

0.189

0.30

-2.23–2.83

LVPW, mm

9.6[8.1–10.8]

9.5[9.0–11.1]

0.64

0.743

0.04

-2.74–2.83

E velocity, cm/sec

90.2[64.5–99.2]

93.8[65.5–98.8]

0.98

<0.001

2.44

-3.38–8.26

A velocity, cm/sec

62.9[40.7–71.3]

64.0[41.4–73.5]

0.98

0.021

1.52

-5.09–8.13

DT, msec

190[165–220]

187[173–211]

0.75

0.638

1.93

-44.3–48.1

e’ velocity (sep), cm/sec

7.3[5.2–9.0]

7.9[6.0–9.2]

0.97

<0.001

0.51

-0.52–1.54

a’ velocity (sep), cm/sec

9.2[7.7–10.2]

9.4[7.8–10.2]

0.95

<0.001

0.33

-9.43–1.09

e’ velocity (lat), cm/sec

8.2[6.8–10.3]

9.2[7.2–11.0]

0.96

<0.001

0.64

-1.10–2.38

a’ velocity (lat), cm/sec

9.3[8.7–11.7]

9.7[8.9–11.5]

0.88

0.117

-0.06

-2.96–2.85

LVOT VTI, cm

16.7[15.3–21.1]

17.1[15.6–21.0]

0.91

0083

0.34

-1.78–2.44

LVOT peak velocity, cm/sec

78.5[74.3–90.3]

80.0[75.3–94.3]

0.93

<0.001

3.03

-5.15–11.2

Abbreviations: LVDd, left ventricular end-diastolic diameter; LVDs, left ventricular end-systolic diameter; IVC, interventricular septum; LVPW, left ventricular posterior wall; DT, deceleration time; LVOT, left ventricular outflow tract.

  1. Comparative Analysis: While the study focuses on AI's accuracy and efficiency in automatic measurements, a comparison with other AI systems or algorithms would provide valuable insights into the relative performance of the system used in this study.

We thank Reviewer #2 for the comment. To our knowledge, no similar studies have examined the accuracy and examination time of AI in other vendors' echocardiography systems in clinical practice. Therefore, we have added the following text to the study limitations section (line 377): ‘This study has not been compared with AI systems from other vendors, nor has it examined the superiority of our system's AI relative to others, warranting further investigation.’

  1. Clinical Impact Discussion: The discussion could be expanded to explore the potential clinical implications of implementing AI in routine TTE measurements more comprehensively. Specifically, it would be helpful to consider the impact on diagnostic timelines and the potential for AI to support decision-making in clinical settings.

We thank the reviewer for the insightful comment. According to your comment, we have added the following sentence to the Discussion section (line 364): ‘While the current study focused on the impact of AI on the accuracy of echocardiographic measurements and examination time, future studies should investigate how AI measurements can influence diagnostic flow and decision-making in clinical practice.’